# A Systematic Review of Joint Spatial and Spatiotemporal Models in Health Research

**DOI:** 10.3390/ijerph20075295

**Published:** 2023-03-28

**Authors:** Getayeneh Antehunegn Tesema, Zemenu Tadesse Tessema, Stephane Heritier, Rob G. Stirling, Arul Earnest

**Affiliations:** 1School of Public Health and Preventive Medicine, Monash University, Melbourne, VIC 3004, Australia; 2Department of Epidemiology and Biostatistics, Institute of Public Health, College of Medicine and Health Sciences, University of Gondar, Gondar 196, Ethiopia; 3Department of Respiratory Medicine, Alfred Health, Melbourne, VIC 3004, Australia; 4Faculty of Medicine, Nursing and Health Sciences, Central Clinical School, Monash University, Melbourne, VIC 3004, Australia

**Keywords:** spatial analysis, joint spatiotemporal analysis, systematic review, public health, geographic information system, disease mapping, shared component models

## Abstract

With the advancement of spatial analysis approaches, methodological research addressing the technical and statistical issues related to joint spatial and spatiotemporal models has increased. Despite the benefits of spatial modelling of several interrelated outcomes simultaneously, there has been no published systematic review on this topic, specifically when such models would be useful. This systematic review therefore aimed at reviewing health research published using joint spatial and spatiotemporal models. A systematic search of published studies that applied joint spatial and spatiotemporal models was performed using six electronic databases without geographic restriction. A search with the developed search terms yielded 4077 studies, from which 43 studies were included for the systematic review, including 15 studies focused on infectious diseases and 11 on cancer. Most of the studies (81.40%) were performed based on the Bayesian framework. Different joint spatial and spatiotemporal models were applied based on the nature of the data, population size, the incidence of outcomes, and assumptions. This review found that when the outcome is rare or the population is small, joint spatial and spatiotemporal models provide better performance by borrowing strength from related health outcomes which have a higher prevalence. A framework for the design, analysis, and reporting of such studies is also needed.

## 1. Introduction

Recent advances in geographic information systems in medicine have led to the development of advanced spatial analysis of geocoded data in health research [1]. Disease mapping, also defined as the spatial analysis of disease risk, is an important area of public health research [2]. Commonly georeferenced data used in epidemiological investigations have information about space and perhaps time [3]. In spatial epidemiology, Tobler’s first law of geography is considered as the foundation of spatial statistics, which asserts that everything is related to everything else, although proximate things are more closely related than distant things [4]. Nearby areas are more likely to share similar geographic characteristics linked to the disease and, likewise, the temporal dependence is greater for succeeding years than for years apart from one another [5,6].

Spatial models have been in use in the field of public health research for decades [7], and important progress over decades has enabled the development of complex models to examine a potential correlation between disease patterns and covariates that are geographically and/or temporally varied [8]. In disease risk mapping, the Standardised Incidence Rate (SIR) and Standardised Mortality Ratio (SMR) are frequently used to measure spatial risk [9]. However, SIR and SMR have limitations when the outcome is rare or when the population is small [10]. To overcome these issues, Bayesian spatial models are applied to obtain smoothed risk by considering spatial dependence (structured and unstructured spatial random effects) in the model [11,12]. Any overdispersion or spatial dependency in the data that cannot be accounted for by the covariates is taken into account by the random effects in the model [13].

Conditional Autoregressive (CAR) and Simultaneous Autoregressive (SAR) prior distributions are routinely used to model the spatially structured random effects [13,14]. In the majority of spatial studies, the spatially structured random effect is modelled using the CAR prior distribution. There are four classes of CAR prior distribution including intrinsic, convolution, Cressie, and Leroux [15]. As mentioned above, data sparseness and a small population are the common shortcomings in spatial modelling. To overcome these limitations, the Besag York Models (BYM) were introduced by Besag et al., 1991 [16]. These models borrow strength from nearby locations and apply spatial smoothing to the risks of the disease, which could improve the accuracy of risk estimations in areas with limited cases or small populations [17].

Rapid advancement in geographically indexed data and statistical innovations has contributed to the growth of spatial studies [18]. The univariate disease mapping approaches have recently been extended to joint disease mapping (modelling multiple interrelated diseases simultaneously) in space and/or time [19,20]. When the desired outcome is rare, the joint spatial models can improve the statistical power by borrowing strength from neighbourhood areas, periods, and/or related highly prevalent outcomes [21]. As many diseases are interrelated and many public health interventions are planned at several dimensions, joint spatial and spatiotemporal analyses are essential for better decision-making and evaluation of already implemented initiatives [22].

The single disease has been studied using univariate spatial and spatiotemporal models; however, these models are not capable of borrowing strength from related diseases [23]. Models that take into account the correlation between diseases improve the estimates of disease risk [24]. The joint spatial and spatiotemporal models combine information from different diseases that share similar risk factors [25]. The majority of spatial and spatiotemporal studies to date have been at the univariate level, considering spatial modelling of specific diseases. However, as many diseases share similar risk factors [26,27,28], applying models that can incorporate data from related diseases is useful from both an epidemiological and statistical perspective [29]. A new field of spatial analysis called shared component spatial and spatiotemporal models [30] analyses both the specific temporal and spatial patterns for each outcome as well as shared spatial and temporal patterns common to multiple outcomes [31].

Joint spatial and spatiotemporal models, as opposed to univariate models, concurrently account for specific and common spatial and temporal effects by incorporating shared spatial and temporal terms [32]. The joint spatial and spatiotemporal models were frequently applied to non-communicable diseases (NCDs) including cancer and diabetes mellitus (DM) [32,33,34,35,36,37,38,39,40,41,42]. The development of joint spatial and spatiotemporal methodologies has coincided with a huge advancement in statistical approaches addressing technical and statistical issues related to advanced spatial statistics. For example, authors employed different Bayesian inference techniques such as Markov chain Monte Carlo (MCMC) [32,33,35,36,37,38,39,40,41,43,44,45,46,47,48,49,50,51,52,53,54,55,56,57,58,59,60] or Integrated Nested Laplace Approximation (INLA) methods [42,43,61,62,63,64,65] in Bayesian shared component spatial and spatiotemporal models.

Although spatial analysis of multiple health outcomes simultaneously has increased over the past few years, a systematic review of published research using joint spatial and spatiotemporal approaches has not yet been undertaken. These studies are heterogeneous in types of analytical models, methodological gaps, spatial and temporal structures, methods of inference, etc. For researchers, especially in the area of spatial statistics, a summary and description of joint spatial and spatiotemporal analysis methods, software, methodological gaps, and modelling concerns are essential.

Therefore, we conducted a systematic review of joint spatial and spatiotemporal models applied to health outcomes to provide insightful recommendations for future researchers as to when and how to fit a joint spatial and spatiotemporal model. This systematic review helps to improve the decision-making process through the joint spatial and spatiotemporal modelling of two or more health outcomes. In addition, the results could inform researchers in terms of providing insights about advanced joint spatial and spatiotemporal statistical methods and related issues.

## 2. Materials and Methods

### 2.1. Data Source and Search Strategy

We performed a systematic review of peer-reviewed published health research that employed joint spatial and spatiotemporal methods. For the formulation of the systematic review methodology, we used the Preferred Reporting Items for Systematic Reviews and Meta-analysis (PRISMA) checklist [66]. We registered the systematic review on the PROSPERO international prospective register of systematic reviews (registration number: CRD42022365445). A comprehensive search strategy was carried out for joint spatial and spatiotemporal models (joint spatial autocorrelation, joint spatiotemporal autocorrelation, joint spatial model, and joint spatiotemporal models) applied to any health or health-related outcomes with no geographic limits.

In our review, a spatial model incorporates a geo-spatial index, a temporal model includes a time index, a spatiotemporal model involves both a geospatial and time index, and joint spatial and spatiotemporal models accommodate a geospatial and/or time index of two or more health outcomes [67,68]. The search was conducted on 19 September 2022. Databases such as PubMed, Medline, Scopus, PsycINFO, Emcare, and Embase were searched. The reference lists of retrieved studies were further searched on Google Scholar and advanced Google to identify more papers. Search terms for the joint spatial and spatiotemporal studies are detailed in Appendix A. Studies published between January 2011 and October 2022 without geographic restrictions were considered in our review. 

Retrieved articles from each database were exported to Endnote version 20 reference citation software and stored as a single file name and then exported to Covidence for further processing (Covidence systematic review software (Veritas Health Innovation, Melbourne, Australia. Available at www.covidence.org, accessed on 20 August 2022)). Duplicates were deleted manually during the title and abstract and full-text screening and automatically in Endnote and Covidence software. Searches were conducted using the following terms: “multivariate spatiotemporal” OR “bivariate spatiotemporal” OR “multivariate spatio-temporal” OR “bivariate spatio-temporal” OR “joint shared spatial model*” OR “joint space-time model” OR “multivariate space-time model*” OR “bivariate space-time model” OR “small area analys*” AND “shared component model” OR “disease mapping” AND “shared component model” OR “space-time mixture model” OR “shared component model” OR “spatial analys*” AND “joint model*” OR “joint spatial model*” OR “joint spatial analys*” OR “shared latent component model” OR “joint model*” AND “spatial model*” OR “spatial factor analys*” OR “risk map*” AND “shared component model*” OR “shared spatial model*” OR “multivariate spatial analys*” OR “bivariate spatial analys*” OR “bivariate conditional autoregressive model” OR “multivariate conditional autoregressive model” OR “joint conditional autoregressive model” OR “joint spatial autocorrelation” OR “bivariate spatial autocorrelation” OR “multivariate spatial autocorrelation” OR “spatial co-cluster*” OR “spatio-temporal co-cluster*” (Appendix A).

### 2.2. Inclusion and Exclusion Criteria

This systematic review covered peer-reviewed studies published in the English language between 2011 and 2022 modelled using joint spatial and spatiotemporal models. The year 2011 was chosen as the starting point because the joint spatial and spatiotemporal analysis of two or more health outcomes was widely implemented as a new area of study over the past decades and because of the need for most recent evidence due to the rapid changes in analytical techniques due to the ongoing advancement in science and technology. Studies before that date were either outdated or superseded by newer methods included in our review. Every article retrieved from the databases (Medline, PubMed, PsycINFO, Emcare, Scopus, and Embase) was exported to Endnote. After excluding duplicates from Endnote, we transferred to Covidence for additional article screening and extraction.

Title and abstracts and full-texts were screened by two authors (GAT and ZTT) independently to identify eligible studies based on the inclusion and exclusion criteria. When conflicts emerged over the inclusion or exclusion of studies, a consensus was reached through discussion, and if the conflicts were not resolved a third reviewer (AE) was consulted. Research articles performed their analyses using the joint spatial approaches; joint spatial and spatiotemporal autocorrelations, joint spatial models, and joint spatiotemporal models, and the outcomes analysed (could be on health outcomes among humans (not animal study)) were eligible for this review. No exclusion was made based on geography and types of health outcomes studied. Conference abstracts, reviews, texts published in a language other than English, non-human studies, and those not considering joint spatial and spatiotemporal models were considered as exclusion criteria.

Methods for spatially and temporally modelling two or more health outcomes, or the same health outcome in two or more subsets of the population at risk, are referred to as joint spatial and spatiotemporal methods [69].

### 2.3. Data Extraction

A data extraction template was developed in Microsoft Excel. The tool was developed considering the review question. Two authors (GAT and ZTT) extracted the data independently. When a disagreement appeared, it was resolved by a third author (AE). The data abstraction tools contained key information such as bibliographic information, research study objectives, the nature and type of data, covariate type, and data analysis methods, i.e., modelling approaches, key findings, and methodological gaps. 

For each study, data such as the last name of the author, article title, name of the journal, year of publication, country, data source, spatial data type, outcomes of interest, number of outcomes, incidence/prevalence of outcomes, inference approaches, estimation techniques, study design, spatial unit, number of the spatial unit, temporal unit, number of temporal units, objectives, sample size, spatial model type, spatial structure, temporal structure, space-time interaction term, assigned priors, the reason for using joint spatial modelling, covariates used in the model, variable selection approach, number of covariate considered in the model, standardisation, spatial neighbourhood structure, temporal adjacency, software used, analysis method, model validation, model comparison measures, effect measure reported, key findings, map reported, script provided, and methodological gaps were extracted. 

### 2.4. Risk of Bias Assessment

Two authors carried out a thorough assessment of the included studies’ methodological quality (GAT and ZTT). All included studies’ risk of bias was evaluated using a quality assessment tool that has an 8-point scoring system updated and modified to evaluate each study’s quality based on its aims and objectives, model validity, overall results, and study conclusion [70,71]. A standardised item list was used to grade the quality and risk of bias of included studies (Appendix A). The checklist consists of 8 questions with possible answers ranging from 0 to 2, with a maximum overall score of 16. Low-, medium-, high-, and very high-quality levels were used to classify the overall score (low = score < 8, medium = score 8–10, high = score 11–13, and very high >13). Two authors (GAT and ZTT) independently assessed each study to determine its score and to determine the overall quality of the included studies. Discussion between the two authors attempted to settle any differences, and for those that could not be settled, a third reviewer (AE) was engaged.

### 2.5. Data Synthesis and Analysis

Microsoft Excel and STATA version 17 software were used for data entry and analysis, respectively. The review findings were summarised into texts, tables, and figures. The descriptive analysis was presented using the proportion, mean, medians, and ranges. 

## 3. Results

### 3.1. Search Results and Characteristics of Included Studies

A comprehensive search of international peer-reviewed journals yielded a total of 4077 published articles. Of these, 4071 articles (PubMed: 2813, Embase: 103, PsycINFO: 24, Emcare: 45, Medline: 82, and Scopus: 1004) were obtained from 6 databases, and 6 more studies were found by manual searches in Google and Google Scholar. All these articles were published from 2011 onwards. During title and abstract screening, about 168 duplicates, 1278 non-relevant studies, and 2544 non-joint spatial studies were discarded, and only 87 studies were left for the full-text screening. A total of 44 studies were excluded from the 87 articles that met the inclusion and exclusion criteria for full-text screening; of these, 32 were not joint spatial models, 4 were methodological reviews, 3 were multivariable analyses, 2 were animal studies, 1 article was a duplication, 1 was a genetic study, and 1 was not a spatial study. The systematic review included 43 research publications that met the eligibility criteria (Appendix A and Figure 1).

Since 2011, the number of publications has fluctuated; it increased in 2016 and reached a peak in 2019 and 2020, then sharply dropped in 2021 before rising again in 2022. More ongoing research in this area is also anticipated with an increase in the number of clinical registries being set up. The majority of studies (n = 30, 69.78%) have been published after 2016 (Figure 2). Nearly one-fifth (18.6%) and 9.3% of the studies were conducted in Iran and the United States of America (USA), respectively (Figure 3). Of the 43 total studies, 15 (34.88%) were applied to infectious diseases, such as HIV/AIDS, herpes simplex virus-2, malaria, Zika, Leishmania, and hookworm, and 11 (25.58%) were applied to cancer, respectively. The *International Journal of Environmental Research and Public Health* (16.67%) and *Spatial and Spatio-temporal Epidemiology* (11.63%) were the most common journals where the articles were published (Table 1). 

### 3.2. Data Source, Study Design, and Unit of Analysis

Seven studies (26.6%) used data from a national health survey or the Demographic and Health Survey (DHS) [43,46,51,60,64,72,75], while seven other studies (26.6%) used data from cancer registries. An estimated 11.63% of the studies used data from multiple surveys. More than one-third (41.86%) and 14 articles (32.56%) were ecological and cross-sectional studies, respectively. The majority (55.81%) of the studies analysed two outcomes simultaneously in the model. Twenty-one studies reported the prevalence/incidence of outcomes, and nearly half of them had a prevalence of less than 10%. 

Given that spatial analysis can be conducted at different spatial scales, about 11 studies performed analyses at the provincial level, and seven studies performed analyses at the county level. The mean number of spatial units was 428, ranging from 11 to 3577 spatial units. Among 43 joint spatial studies considered for the systematic review, 17 employed a joint spatiotemporal model. Of them, the vast majority (n = 15, 88.23%) used year as the unit of analysis. The mean temporal period was 21.71, ranging from 5 to 260 (Table 2). 

### 3.3. Spatial Data and Modelling Techniques

Only seven studies (n = 7, 16.28%) used point data, while the majority of the studies (n = 36, 83.72%) used areal data aggregated at a specific geographic unit, such as a municipality, province, county, Statistical Local Area (SLA), state, etc. to estimate the diseases’ relative risks [59,61,62,76,77,78,79]. The Bayesian estimation approach was used in more than three-quarters (n = 35, 81.40%) of the included studies [32,33,35,36,37,38,39,40,41,42,43,44,45,46,47,48,49,50,51,52,53,54,55,56,57,58,59,60,61,62,63,64,65,73,79]. For Bayesian inference, the MCMC estimation approach was utilised in 27 studies [32,33,35,36,37,38,39,40,41,43,44,45,46,47,48,49,50,51,52,53,54,55,56,57,58,59,60], and for seven studies (16.28%), INLA was employed [42,43,61,62,63,64,65]. 

To analyse two or more health outcomes simultaneously, several joint spatial and spatiotemporal models were used. To investigate the relative risk of the study variables and their risk factors, joint spatial models were used in 24 studies [36,38,39,40,42,43,45,46,47,48,49,50,51,52,54,56,57,59,60,61,62,64,73,75], joint spatiotemporal models in 12 studies [32,33,35,37,41,44,55,58,63,65,80], and joint spatial and spatiotemporal autocorrelation methods such as Moran Index statistics, Local Indicator Spatial Analysis (LISA), Getis Ord Gi statistics, or Kulldroff spatial and Spatio-temporal scan statistical tests used in 7 studies [34,72,74,76,77,78,79].

In joint spatial temporal models, the structured and unstructured spatial random effects were considered to account for the spatial dependence and independent effects, respectively. In more than two-thirds (n = 26, 72.2%) of the studies, prior CAR was considered for the spatially structured random effect [32,33,35,36,37,38,39,40,41,43,44,46,47,48,49,50,51,52,53,55,58,60,61,63,64,65], and all assigned Identical and Independent Distribution (IID) for the unstructured spatial random effects. Out of 12 joint spatiotemporal studies, seven (58.23%) considered prior first-order random walk to account for the temporal dependence in the model [32,33,35,41,53,58,65]. The simple exchangeable hierarchical structure was taken into consideration for the spatiotemporal interaction terms in five of the joint spatial and temporal studies [32,33,41,53,65]. 

Of the 43 studies, 26 studies used R software [34,35,36,42,43,44,45,48,51,52,54,55,56,57,58,59,60,61,62,63,64,65,75,77,79,80], 21 studies used either WinBUGS or OpenBUGS or GeoBUGS [32,33,36,37,38,39,40,41,43,44,46,49,50,51,53,54,56,63,73,78,79], 7 studies used ArcGIS or QGIS [38,41,50,53,60,75,77], 4 studies used GeoDa [34,45,72,74], 3 studies used for SaTScan [76,77,80], and 2 studies used Fortran software [47,78]. Numerous joint spatial and spatiotemporal statistical techniques were used to examine spatial risk factors. Seven studies were joint spatial/spatiotemporal autocorrelation studies [34,48,72,74,76,77,80], the joint Bayesian shared Spatiotemporal model was used in four studies [42,53,60,65], and the multivariate Bayesian Spatiotemporal shared component model with Poisson distribution was used in two studies [33,37]. Other models included the bivariate Bayesian logit spatial model [46,51,63,64], geo-additive mixed models [57,75], and the multivariate negative binomial models with CAR random effects [43,80] (Table 3).

### 3.4. Covariates, Model Validation, and Goodness of Fit Assessment

Different measures of joint spatial and spatiotemporal model performance were reported. The majority (n = 22, 51.16%) of the studies considered Deviance Information Criteria (DIC) for model comparison [32,33,37,38,39,41,43,44,46,47,48,49,52,53,54,60,61,62,63,64,73,78], and six studies used Root Mean Predictive Squared Error (RMPSE) [35,37,44,58,59,78]. A combination of model comparison measures was used in many of the studies. The common effect measures reported in the included studies were relative risk (n = 17, 47.22%), odds ratio (n = 9, 25%), and coefficients (n = 8, 22.22%). Very limited studies underwent model validation (n = 5, 13.89%) [42,58,59,63,80]. 

Almost all joint spatial and spatiotemporal studies used maps to present the risk estimates. In the joint spatial and joint spatiotemporal models, socio-economic variables were the predominant variables among the covariates considered in the model to predict the outcomes across space or space-time [34,37,38,43,46,47,48,52,54,55,57,59,60,64,73,75]. Regarding standardisation, only five of the joint spatial and spatiotemporal models applied standardisation for common demographic variables [42,58,59,63,80]. Queen contiguity was the most commonly used method to define the neighbourhood structure (n = 10, 23.26%) [45,48,50,53,54,55,60,65,72,75] (Table 3).

### 3.5. Key Implications of Applying Joint Spatial Modelling, Findings, and Methodological Gaps

The justifications provided in the included studies for fitting the joint spatial and spatiotemporal model, shared component spatial and spatiotemporal model, or multivariate spatial and spatiotemporal model varied. Fourteen (38.89%) studies applied the shared component spatial and spatiotemporal model to consider the spatial dependence of interrelated outcome variables and to better explore their overlapping epidemiology [43,44,47,52,57,58,60,61,62,65,72,73,75,78]. Of the 36 joint spatial and spatiotemporal studies, 12 (33.33%) studies used the joint model for the ease of interpretation and to improve the precision of estimation [38,39,43,44,49,53,54,55,57,60,65,76]. Different reasons were provided for using the joint spatial and spatiotemporal analysis. Nine studies (25%) applied the joint spatial and spatiotemporal model to borrow strength between diseases and to incorporate data from a more common and related disease when interest was in a relatively rare disease, thereby strengthening the relevant results of the rare disease [36,37,42,44,49,53,55,60,61].

Out of 43 studies, 31 studies (72%) found reasonable patterns in the co-occurrence of health outcomes in geographic prevalence across areas [32,34,35,38,40,41,42,45,46,47,48,49,50,51,52,54,55,56,57,58,59,61,64,65,72,74,75,76,77,79,80]. The joint spatial and spatiotemporal model yields more precise and efficient estimates, especially when the number of observed events is rare [33,43,60,62,73,78]. Besides, the shared component joint spatial model had a better model fit relative to a joint spatial model without the shared component [36,46,53,62,73].

The studies included in this systematic review have self-reported methodological gaps. Seven studies acknowledged that in aggregated data, ecological fallacies are introduced, and some relevant information may be concealed by using large geographical units of study [40,47,48,49,53,65,80]. Therefore, using smaller units of analysis as a methodological gap may be a preferred approach. Four of the studies revealed that a meaningful number of temporal units is required to efficiently detect the temporal effect [38,41,44,80], and assuming the shared and specific components as independent ignores the possibility of interactions between the true covariates [38,44,62,73]. Three of the studies reported that MCMC has computational problems, model fitting, and convergence issues [42,43,56] (Table 4).

### 3.6. Assessment of Quality

Using the adapted quality assessment tool of modelling study qualities, the quality scores ranged from 0 to 16. The median quality score was 12/16, ranging from 8 to 16. Ten studies were classified as medium quality, twenty-one studies as high quality, and 12 studies as very high quality (Table 5).

## 4. Discussion

In this study, joint spatial and spatiotemporal models in health research were systematically reviewed. These models were mainly applied to infectious diseases [54,57,59,60,61,62,63,64,65,75,76,77,78,79,80], cancer [32,33,34,35,36,37,38,39,40,41,42], chronic diseases [44,48,51,58,72,73,74], and maternal and child health outcomes [46,52,53]. This showed that infectious diseases, which have a major worldwide burden and have the potential to spread to nearby areas, are currently receiving significant attention from researchers [81,82]. The spatial and spatiotemporal studies of interrelated infectious diseases provide a better understanding of the magnitude, pattern, overlapping epidemiology, and shared individual and area-level risk factors. The majority of infectious diseases co-occur in the same patients such as tuberculosis with HIV, herpes simplex virus-2 with HIV, leishmaniasis with malaria, etc. [83]. 

Joint spatial and spatiotemporal modelling of two or more cancers has become increasingly frequent over the past few decades [24,29,84] to examine the shared and differing trends of cancers regarding geographic patterns and shared risk factors. Contrary to univariate analysis, joint spatial models include shared components as various groups of cancers share common risk factors [85,86]. The majority of studies in spatial studies were based on single health outcomes, even though diseases such as cancer have common risk factors. Joint spatial and spatiotemporal modelling has recently become popular. It can model rare and common cancers to improve the estimates by borrowing strengths. It is feasible to modify behaviour common to cancers, and this has huge potential for preventing cancer. Major cancer risk factors can be altered by applying behavioural strategies including ceasing smoking, getting more exercise, managing weight, improving diet, limiting alcohol consumption, getting regular cancer screening tests, and limiting sun exposure [87]. Given that many of these cancer-preventive techniques lower the risk of many cancers, these might be supported by generating evidence through shared spatial and spatiotemporal studies.

The number of publications on joint spatial studies decreased dramatically during the coronavirus disease (COVID-19) period. It could be explained by the fact that since 2019, journals have prioritised COVID-19 research since little was known about the disease [88]. Evidence on the route of transmission, typical clinical features, underlying risk factors, and pathogenesis was limited, which is why the spatial analysis of COVID-19 with other infectious diseases was infrequent. In addition, advanced statistical approaches such as joint spatial and spatiotemporal models were not commonly applied as little was known about COVID-19’s clinical manifestation, route of transmission, etiology, and treatment. 

More than one-third of the studies used ecological data for analysis. This demonstrates how disease mapping frequently used aggregated data at a specific geographic level to generate area-level estimates to guide healthcare decisions and effective resource allocation [89]. Compared with other studies, ecological data can be accessed or retrieved from reports quite easily. The main data sources were DHS or national health survey data [43,46,51,60,64,72,75] and cancer registry data [35,36,37,38,39,41,42]. This could be because of DHS and other national health surveys having location data (GPS) and geolocated covariates including environmental, pollution, demographic, and socio-economic covariates available in these surveys [90]. 

Some of the studies were exploratory spatial analyses, including joint spatial autocorrelation techniques such as Moran Index (MI) statistics, Local Indicator Spatial Analysis (LISA), Getis Ord Gi^*^ statistics, or Kulldroff spatial or spatiotemporal scan statistical tests [34,72,74,76,77,78,79]. Unlike univariate spatial autocorrelations, multivariate spatial autocorrelations can determine how interrelated health outcomes such as TB and HIV, HSV-2 and HIV, and cancers influence each other spatially or spatiotemporally. They can explore the overlapping spatial and/or temporal distribution of two or more interrelated diseases. To detect the local clusters (hotspot and cold spot areas), spatial and space–time scanning statistical analysis and Getis-Ord Gi statistic are commonly used for cluster detections [91]. The hotspot and coldspot cluster detection is sensitive to the change in the size of the spatial and temporal units of analysis in which the data are aggregated; thus, analysing at a small spatial scale is preferable to identify hotspot areas efficiently. Multivariate spatial autocorrelation methods can investigate the spatial dependence of two or more interrelated health outcomes. In multivariate spatial autocorrelation analysis, the presence of a disease in a particular area is not only influenced by the prevalence of diseases in neighbouring areas, but also influenced by the presence of diseases that are related to one another in the area. However, they are unable to look at how the existence of one health outcome in one area may affect the spread of someone else in nearby areas, how one health outcome may affect the spread of another in adjacent areas, or, further, how it is affected by the spatial risk factors [92].

When variables were accounted for, the majority of the joint spatial and spatiotemporal models showed model improvement [32,34,37,38,43,45,46,47,48,51,52,53,54,55,57,59,60,61,64,73,75,79,80]. If the covariates were considered, joint spatial and spatiotemporal models performed much better than others and offered more insights than univariate models [93]. When the outcome is rare, the joint spatial models could improve the model performance by borrowing strength from interrelated diseases, neighbourhood areas, and/or time [10]. Besides, they can capture the spatial and spatiotemporal effects unexplained by the observed covariates by introducing random effects. This is well-suited to the data that have limited predictors and models that capture few covariates.

Only five studies underwent standardisation for common demographic variables [38,40,48,55,61]. Demographic covariates such as age, sex, race, etc. are the most obvious risk factors for almost all health and health-related conditions. In the univariate spatial analysis, the difference in incidence or mortality of given diseases because of age, sex, or other variables can be addressed by standardisation, which ignores the effects of these variables in the analysis. Age, sex, and other demographic variables are determinants for a multitude of health-related and other outcomes. The impacts of age, sex, and other common risk factors should be estimated using these variables in the model, as well as their interactions with other factors, such as spatial and temporal effects. 

The vast majority of the studies were conducted based on a Bayesian modelling framework [32,33,35,36,37,38,39,40,41,42,43,44,45,46,47,48,49,50,51,52,53,54,55,56,57,58,59,60,61,62,63,64,65,73,79]. This was in line with the advancement of Bayesian statistics in the field of disease mapping. Bayesian spatial models are now commonly applied because of advances in knowledge of advanced statistics, programming, coding skills, and improved computer power resources that overcome computational problems. This methodology can provide more reliable area-level estimates specifically when the cases are rare or the population is small [90]. It can smoothen the observed extreme estimates towards the global mean or the neighbourhood values and provide more robust estimates, specifically when studied areas have sparse populations, by assigning prior distributions to define spatial structure to ensure the closer areas have more contribution than distant areas [22,94].

The majority of the Bayesian spatial and spatiotemporal studies undertook model fitting and inference based on MCMC [32,33,35,36,37,38,39,40,41,43,44,45,46,47,48,49,50,51,52,53,54,55,56,57,58,59,60] followed by INLA [42,43,61,62,63,64,65]. The most popular models for spatial studies were Bayesian hierarchical models with structured and unstructured random effects. The MCMC approach is computationally very demanding and has convergence issues [95]. This is especially true for hierarchical models, which by their very nature make MCMC convergence unpredictable and slow. No matter the model, it is necessary to assess the convergence of posterior samples because there is no guarantee that such models can be easily fitted, in which additional simulations and model simplification will be necessary. In contrast, INLA has recently emerged as a reliable alternative method for fitting Bayesian spatial models that overcomes MCMC’s drawbacks [96,97]. The package uses INLA to estimate Bayesian models without the need for posterior sampling techniques. In practical terms, numerical integration is performed via this approximation; therefore, it does not require a lot of iterative processing [98]. Bayesian estimation utilising the INLA methodology typically takes substantially less time than MCMC, which is the reason why this package was developed for spatial statistics in the first place.

The majority of studies applied joint spatial models [36,38,39,40,42,43,45,46,47,48,49,50,51,52,54,56,57,59,60,61,62,64,73,75] followed by joint spatiotemporal models [32,33,35,37,41,44,55,58,63,65,80]. Unlike spatial autocorrelation studies, the spatial and spatiotemporal models can investigate the covariates influencing the distribution of the outcome across space and/or time [33,37,42,43,53,57,60,65,75,80]. The joint spatial and spatiotemporal models are applied for spatial analysis of two or more health outcomes or one outcome across different population groups. They typically rely on Generalised Linear Mixed Models (GLMMs) that consider shared and specific spatial, temporal, and spatiotemporal random effects. The data type, distribution, nature, and incidence of the outcomes determine the type of joint spatial and spatiotemporal models, e.g., poisson or negative binomial GLMM for count data and logistic regression for categorical outcomes. The dependence between diseases with similar spatial or temporal patterns is captured by prior distributions [13]. The multivariate CAR models smoothen noisy estimates and leverage information from nearby areas and interrelated diseases to predict spatially autocorrelated area-level disease risks [99]. However, multivariate CAR models are unable to show how the correlation of outcomes varies across space. The Copula geoadditive model overcomes this limitation and can demonstrate the change in the association between outcomes across geographic locations [100]. However, it is unable to detect the geographic areas contributing to higher or lower risk of simultaneous occurrence of multiple outcomes. Recently, the majority of studies applied the shared spatial and spatiotemporal models that decompose the spatial effect into shared and disease-specific spatial effects [24].

The majority of studies used a prior CAR for the spatially structured random effects to account for the spatial dependence [32,33,35,36,37,38,39,40,41,43,44,46,47,48,49,50,51,52,53,55,58,60,61,63,64,65], and all assigned IID for the unstructured spatial random effects. MCAR models incorporate both spatially structured and unstructured random effects in the model. The spatial dependence among adjacent areas is accounted for by assuming a CAR process in the random effects [101]. Most studies used first-order random walk prior to temporal random effects [32,33,35,41,53,58,65].

The authors’ reasons for fitting joint spatial and spatiotemporal models over univariate models were the following: to considering the spatial dependence of related outcomes in the model and better explore their overlapping epidemiology [43,44,47,52,57,58,60,61,62,65,72,73,75,78]; to improve estimation precision [38,39,43,44,49,53,54,55,57,60,65,76]; and to strengthen relationships between diseases by borrowing data from a more prevalent and related disease when the disease of interest is relatively rare [36,37,42,44,49,53,55,60,61]. In addition, the studies revealed that the shared component models yield more precise and efficient estimates, especially when the disease is rare or the population is small [33,43,60,62,73,78]. Moreover, incorporating the shared component in the model could improve the model’s performance [36,46,53,62,73].

This review pointed out several recommendations for the development of improved joint spatial and spatiotemporal models. Some studies acknowledged that when data are aggregated, ecological fallacies are introduced, and some relevant information may be concealed by using large geographical units of study [40,47,48,49,53,65,80]. Thus, using smaller units of analysis increases the precision of the estimates. Assuming the shared and specific components as independent denies the possibility of interactions between the true covariates [38,41,44,80], and a relevant number of temporal units is necessary to efficiently identify the temporal effect [38,44,62,73]. Moreover, some of the studies showed that MCMC has computational problems, model fitting, and convergence issues [42,43,56]. INLA is better for developing statistical models to obtain efficient risk estimations and direct the efficient distribution of medical interventions.

DIC was the most commonly used model comparison criteria to measure and compare the model goodness of fit and model complexity [32,33,37,38,39,41,43,44,46,47,48,49,52,53,54,60,61,62,63,64,73,78], followed by RMSPE [35,37,44,58,59,78]. Most of the studies used a combination of goodness-of-fit measures for model assessment. DIC and WAIC are model performance measures that are calculated by combining the model likelihood function (deviance (-2LLR)) and a model complexity term (number of effective parameters) [102,103]. In addition to model performance assessment, model accuracy assessment such as mean absolute prediction error and mean square prediction error are considered for model comparison [104]. Moreover, local measures of fit such as the conditional predictive ordinate are also used for making a model comparison. Apart from three studies [34,44,77], most studies reported maps for the visualisation of risk estimates. Maps offer epidemiologists enough evidence to display spatial risk and/or risk factors across time and/or space. It can give decision-makers motivation, insight, and assist potential health interventions in high-risk areas.

The scientific community may benefit from the epidemiological and statistical insights this systematic study provides in terms of joint spatial and spatiotemporal model applications in health research. First, the utility of joint spatial and spatiotemporal models is more pronounced in large data registries and when multiple interrelated diseases are fitted simultaneously. It is therefore crucial to estimate the smoothed relative risk of lower-prevalence cases through the borrowing of strength from the related cases and neighbourhood areas. Although there were joint spatial and spatiotemporal studies, the systematic review found heterogeneity in methods of estimation technique, statistical models, prior selections, defining adjacencies, and model complexities. This showed that a consistent framework for undertaking joint spatial and spatiotemporal models is needed. This framework is currently a focus of our research program. This systematic review provides insight suggesting that jointly modelling two or more cases that have shared characteristics is better to detect clusters of cases specifically when the number of cases is rare, such as in rare cancers and orphan diseases, or when the population is small. Another important finding was that the most complex models (joint spatial and spatiotemporal models incorporating covariates and interaction) performed very well. Overall, the systematic review identified several areas of improvement in joint spatial studies such as providing data, maps, scripts, and methodological gaps.

This review has some strengths and limitations, including an extensive search of six electronic databases to retrieve studies in an area without a previous systematic review. Careful title/abstract and full-text screening was carried out with predefined inclusion and exclusion criteria. One of the limitations of this review was that the majority of the studies were from a few countries, which might be because of spatial data being limited (GPS data and software packages for joint spatial and spatiotemporal model) or insufficient funding or statistical skills, indicating the possibility of publication bias or a focus of research effort on countries included in research publications. Another limitation was that only articles published in English were considered, so we may have excluded valuable contributions.

## 5. Conclusions

Multivariate disease mapping is crucial for understanding the burden of interrelated health outcomes over space and/or time. Numerous joint spatial and spatiotemporal methodologies aiming to explore the spatial risk of two or more health outcomes simultaneously were reviewed. The majority of studies used Bayesian methods, which handled a wider range of variance components at different levels in the model and could consider model uncertainties to provide reliable estimates. The most often utilised covariates in joint spatial and spatiotemporal models were socio-economic and demographic. Most of the reviewed studies used shared component spatial and spatiotemporal models with a Poisson-based and negative binomial modelling approach. Relatively few studies have been published on the applications of joint spatial and spatiotemporal models since the COVID-19 pandemic. Reviewed studies have acknowledged that aggregated data are liable to ecological fallacies and some relevant information may be concealed by using large geographical units of study. Therefore, this systematic review highlighted the need for future joint spatial and spatiotemporal models to analyse correlated health outcomes to guide decision-making for effective prevention and control strategies.

## Figures and Tables

**Figure 1 ijerph-20-05295-f001:**
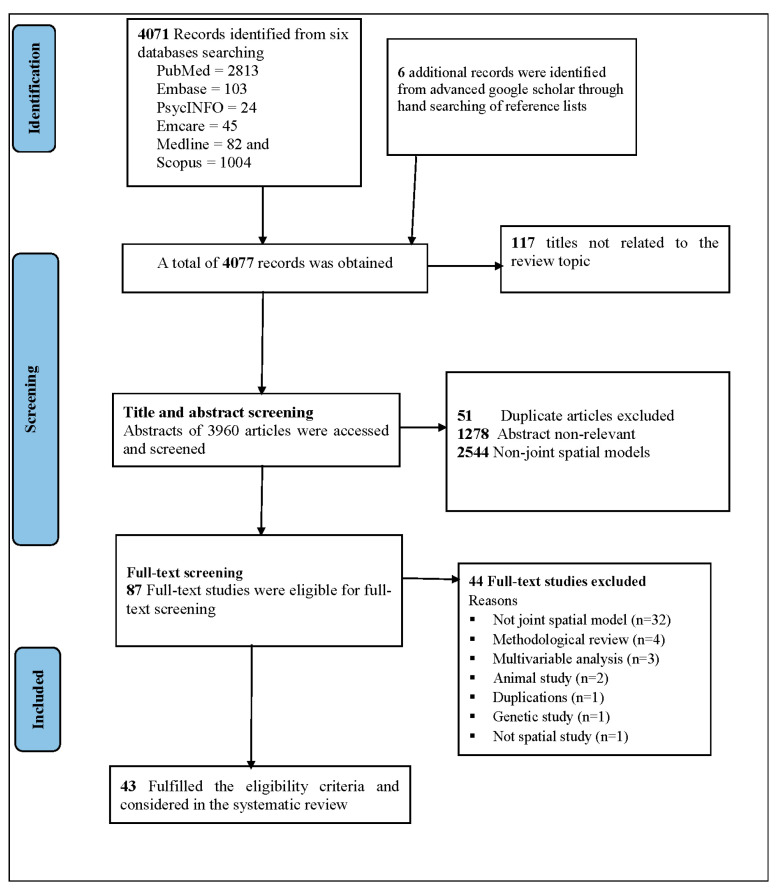
Flow chart of selection of studies for the systematic review using PRISMA checklists from 2011–2022.

**Figure 2 ijerph-20-05295-f002:**
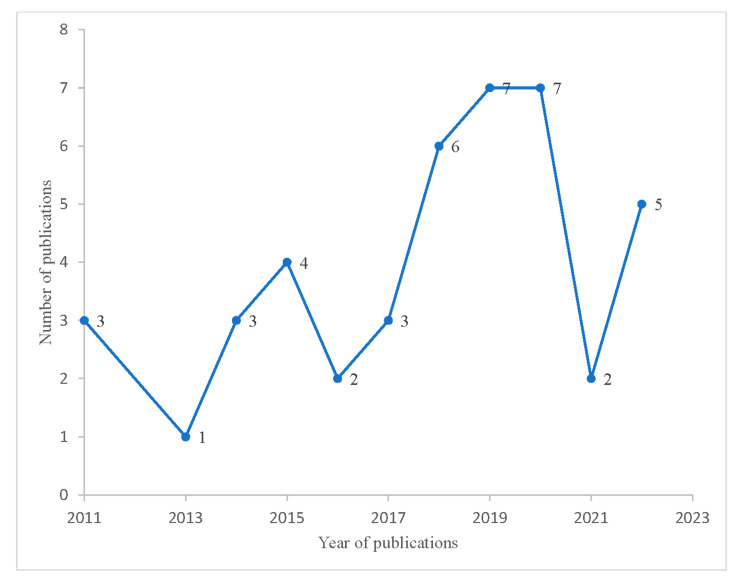
Number of studies based on year of publication from 2011 to 2022.

**Figure 3 ijerph-20-05295-f003:**
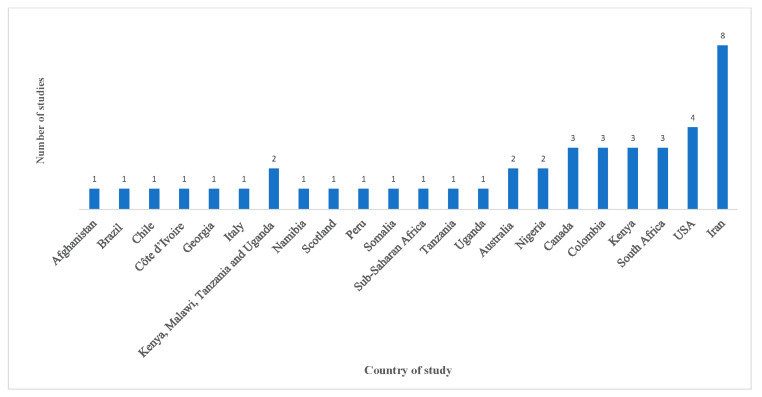
Number of included articles by country of study.

**Table 1 ijerph-20-05295-t001:** General characteristics of the included studies.

Characteristics	Frequency	Percentage (%)	References
Study category			
Cancer	11	25.58	[32,33,34,35,36,37,38,39,40,41,42]
Chronic diseases	7	16.28	[44,48,51,58,72,73,74]
Infectious diseases	15	34.88	[54,57,59,60,61,62,63,64,65,75,76,77,78,79,80]
Health service utilisation	1	2.33	[49]
Maternal and child health outcomes	3	6.98	[46,52,53]
Others *	6	13.95	[43,45,47,50,55,56]
Publication journal			
*International Journal of Environmental Research and Public Health*	7	16.67	[35,40,49,51,64,72,73]
*Spatial and Spatio-temporal Epidemiology*	5	11.63	[46,48,55,57,61]
*PLOS ONE*	3	6.98	[44,54,65]
*Statistics in Medicine*	2	6.65	[58,78]
*Statistical Methods in Medical Research*	2	4.65	[52,63]
*BMC Public Health*	1	2.32	[53]
*Malaria journal*	1	2.32	[79]
*Epidemiology and infection*	1	2.32	[80]
*Annuals of GIS*	1	2.32	[50]
*Geospatial Health*	1	2.32	[74]
*International Journal of Preventive Medicine*	2	4.65	[33,39]
*International Statistical Review*	1	2.32	[62]
*African Health Sciences*	1	2.32	[43]
*Journal of Health, Population, and Nutrition*	1	2.32	[75]
Others **	12	27.91	[32,34,36,37,38,45,47,56,59,60,76,77]

Others *: Trauma, injury, mental health, drug and substance use: Others **: *BMC Paediatrics*, *Acta Tropica*, *The International Journal of Cancer Epidemiology*, *Detection, and Prevention*, *Osong Public Health and Research Perspectives*, *Accident Analysis and Prevention*, *Rev Saude Publica (RSP)*, *Asian Pacific Journal of Cancer Prevention*, *Canadian Journal of Public Health, Biometrics*.

**Table 2 ijerph-20-05295-t002:** Data, study design, and covariates of the reviewed studies.

Item	Category	Number	Percentage (%)	References
Data source(s)	DHS or National health survey	7	16.28	[43,46,51,60,64,72,75]
	Malaria indicator survey	3	6.98	[57,63,75]
	HMIS/DHIS	2	4.65	[53,80]
	Death and cause of death registration system	1	2.33	[33]
	Multiple surveys	5	11.63	[37,38,47,65,80]
	Hospital records	2	4.65	[44,59]
	AIDS indicator survey	1	2.33	[54]
	Cancer registry	7	16.28	[35,36,37,38,39,41,42]
	Others *	23	53.49	[32,40,46,47,48,49,50,51,52,53,54,55,56,58,61,62,63,73,74,76,77,78,79]
Study design (More than one design was applied in some of the studies)	Ecological	18	41.86	[32,33,34,38,39,40,47,48,49,52,55,58,61,73,74,76,77,79]
	Cross-sectional	14	32.56	[43,46,51,54,56,57,60,62,63,64,72,75,78,80]
	Retrospective	9	20.93	[35,36,37,41,42,44,45,59,65]
	Longitudinal	2	4.65	[51,53]
	Others **	2	4.65	[49,50]
Number of outcomes of the study	2	24	55.81	[33,34,36,37,40,41,45,46,48,49,50,52,54,55,58,60,61,65,72,73,75,76,78,79,80]
	3	7	16.28	[35,39,43,62,64,74,77]
	4	2	4.65	[51,53]
	5	2	4.65	[44,56]
	6	0	0	----
	7	2	4.65	[32,38]
Prevalence of outcomes of the study	All less than 10%	11	25.58	[33,34,35,36,37,41,42,46,52,53,61]
	Either of them is less than 10%	4	9.30	[48,51,79,80]
	All greater than 10%	6	13.95	[43,45,60,62,73,75]
	Not reported	22	51.16	[32,38,39,40,44,47,49,50,54,55,56,57,58,59,63,65,72,74,76,77,78]
Spatial unit	Provinces	11	25.58	[32,33,35,36,38,39,41,43,45,51,80]
	County	7	16.28	[42,47,48,52,53,54,65]
	Municipalities	4	9.30	[40,74,76,77]
	Districts	3	6.98	[60,72,73,75]
	Schools/Health facility	3	6.98	[34,78,79]
	SLAs	1	2.33	[37]
	Not reported	1	2.33	[59]
	Others ***	12	27.91	[44,46,49,50,55,56,57,58,61,62,63,64]
Temporal units (n = 17)	Year	15	88.23	[32,33,34,35,37,41,44,45,53,58,62,65,74,76]
	Month	1	5.88	[80]
	Weeks	1	5.88	[77]

Others *: Food Security and Nutrition Unit, NASA, ABS, Urban Malaria Control Program (UMCP), Others **: Survey reports, Others ***: residence, constituents, point, cluster, state, ZIP code, dissemination area, intermediate geography, ten cell count, neighbourhood.

**Table 3 ijerph-20-05295-t003:** Details of the types, structures, and methods of joint spatial models of included studies (n = 43).

Items	Number	Percentage (%)	References
Types of spatial data			
Point	7	16.38	[59,61,62,76,77,78,79]
Area	36	83.72	[32,33,34,35,36,37,38,39,40,41,42,43,44,45,46,47,48,49,50,51,52,53,54,55,56,58,60,63,64,65,72,73,74,75]
Methods of inference			
Frequentist	8	18.60	[34,72,74,75,76,77,78,80]
Bayesian	35	81.40	[32,33,35,36,37,38,39,40,41,42,43,44,45,46,47,48,49,50,51,52,53,54,55,56,57,58,59,60,61,62,63,64,65,73,79]
Estimation techniques (n = 36)			
ML	2	4.65	[75,80]
MCMC	27	62.79	[32,33,35,36,37,38,39,40,41,43,44,45,46,47,48,49,50,51,52,53,54,55,56,57,58,59,60]
INLA	7	16.28	[42,43,61,62,63,64,65]
Joint spatial analysis techniques			
Joint spatial autocorrelation analysis	7	16.28	[34,72,74,76,77,78,79]
Joint spatial models	24	55.81	[36,38,39,40,42,43,45,46,47,48,49,50,51,52,54,56,57,59,60,61,62,64,73,75]
Joint Spatio-temporal models	12	27.91	[32,33,35,37,41,44,55,58,63,65,80]
Spatial structure (n = 36)			
MCAR/BCAR/ ICAR/CAR	26	72.22	[32,33,35,36,37,38,39,40,41,43,44,46,47,48,49,50,51,52,53,55,58,60,61,63,64,65]
SAR	1	2.78	[45]
GMRF	3	8.33	[42,54,57]
Not reported	6	16.67	[56,59,62,73,75,80]
Temporal structure (n = 12)			
Prior first-order random walk	7	58.33	[32,33,35,41,53,58,65]
log-linear structure	1	8.33	[44]
Prior first-order autoregressive	2	16.66	[37,55]
Second-order random walk	1	8.33	[63]
Not reported	1	8.33	[80]
Spatio-temporal term (n = 12)			
Uncorrelated ST interaction term	1	8.33	[35]
Simple exchangeable hierarchical Structure	5	41.67	[32,33,41,53,65]
First order autoregressive	1	8.33	[58]
Not reported	5	41.67	[37,44,55,63,80]
The software’s used			
R/R-studio/R2WinBUGS/R-INLA	26	60.47	[34,35,36,42,43,44,45,48,51,52,54,55,56,57,58,59,60,61,62,63,64,65,75,77,79,80]
ArcGIS/QGIS	7	16.28	[38,41,50,53,60,75,77]
WinBUGS/OpenBUGS/GeoBUGS	21	48.84	[32,33,36,37,38,39,40,41,43,44,46,49,50,51,53,54,56,63,73,78,79]
GeoDA	4	9.30	[34,45,72,74]
SaTScan	3	6.98	[76,77,80]
Fortran/MATLAB	2	4.65	[47,78]
Spatial models used (n = 36)			
A multivariate negative binomial model with CAR random effects	2	5.56	[43,80]
Multivariate Bayesian Spatio-temporal shared component model with Poisson distribution	2	5.56	[33,37]
Poisson generalised linear mixed model (GLMM) with a shared spatial component with the log-linear temporal trend	1	2.78	[44]
Multivariate spatial autocorrelation and hotspot analysis	7	19.44	[34,48,72,74,76,77,80]
Joint spatial marked point processes model with Poisson distribution	1	2.78	[61]
Bayesian multivariate ST mixture model	1	2.78	[35]
Bivariate bayesian logit spatial model	4	11.11	[46,51,63,64]
Bayesian hierarchical geostatistical shared component model/ Bivariate bayesian geostatistical logistic model	2	5.56	[62,78]
A bayesian multivariate conditional auto-regressive model with Poisson distribution	1	2.78	[48]
Bayesian spatial Polytomous Logit Model	1	2.78	[39]
Bayesian spatial biprobit model	1	2.78	[52]
Joint bayesian Spatio-temporal shared component binomial model/Bayesian joint hierarchical Spatio-temporal Log-linear model/Bayesian shared component model	4	11.11	[42,53,60,65]
Bayesian semi-parametric spatial joint model/Bayesian nonparametric model using Gaussian processes for the analysis of spatially distributed multivariate binary outcome	2	5.56	[54,59]
Geoadditive mixed model	2	5.56	[57,75]
Bayesian geostatistical shared component multinomial modelling	1	2.78	[78]
Bayesian ANOVA	1	2.78	[56]
Model validation (n = 36)			
No	31	86.11	[32,33,35,36,37,38,39,40,41,43,44,45,46,47,48,49,50,51,52,53,54,55,56,57,60,61,62,64,65,73,75]
Yes	5	13.89	[42,58,59,63,80]
Model comparison metrics (n = 36)			
DIC	22	51.16	[32,33,37,38,39,41,43,44,46,47,48,49,52,53,54,60,61,62,63,64,73,78]
WAIC	4	9.30	[35,40,55,62]
CPO	2	4.65	[37,62]
PIT	1	2.33	[62]
RMSPE/Mean absolute error	6	13.95	[35,37,44,58,59,78]
KL	1	2.33	[40]
Credible interval plot	1	2.33	[78]
Bayesian *p*-value and L-criterion	1	2.33	[37]
Others * (AIC, BIC)	2	4.65	[75,79]
Effect measure reported (n = 36)			
OR	9	25.00	[43,46,60,61,62,63,64,73,79]
RR	17	47.22	[32,33,36,37,38,39,40,41,42,47,48,49,50,53,54,55,65]
Coefficient	8	22.22	[35,44,45,51,52,57,58,59,80]
Covariates (n = 36)			
Demographic	14	38.89	[43,46,47,51,52,54,57,59,60,61,64,73,75,79]
Socio-economical	16	44.44	[34,37,38,43,46,47,48,52,54,55,57,59,60,64,73,75]
Environmental	6	16.67	[37,45,60,75,79,80]
Clinical, health service, and behavioral related	6	16.67	[32,43,46,53,57,73]
Standardisation (n = 36)			
No	31	86.11	[32,33,35,36,37,39,41,42,43,44,45,46,47,49,50,51,52,53,54,56,57,58,59,60,62,63,64,65,73,75,80]
Yes	5	13.89	[38,40,48,55,61]
Method to define spatial neighbourhood structure			
Distance-based neighbourhood matrix	1	2.33	[34]
Queen contiguity	10	23.26	[45,48,50,53,54,55,60,65,72,75]
Rook contiguity	2	4.65	[46,52]
Non-specified adjacency based	3	6.98	[33,44,49]
Not reported	27	62.79	[32,35,36,37,38,39,40,41,42,43,47,51,56,57,58,59,61,62,63,64,73,74,76,77,78,79,80]
Map reported			
No	3	6.98	[34,44,77]
Yes	40	93.02	[32,33,35,36,37,38,39,40,41,42,43,45,46,47,48,49,50,51,52,53,54,55,56,57,58,59,60,61,62,63,64,65,72,73,74,75,76,78,79,80]
Script provided (n = 36)			
No	31	86.11	[32,33,36,37,38,39,40,41,42,43,44,45,46,47,48,49,50,51,52,55,56,57,58,59,60,61,62,64,73,75,80]
Yes	5	13.89	[35,53,54,63,65]

AIC: Akaike Information Criteria, ANOVA: Analysis of Variance, BIC: Bayesian Information Criteria, CPO: Conditional Predictive Ordinate, DIC: Deviance Information Criteria, KL: Kullback Leibler Divergence, OR: Odds Ratio, PIT: Probability Integral Transform, RMSPE: Root Mean square Predictive Error, RR: Relative Risk, WAIC: Watanabe Akaike Information Criteria.

**Table 4 ijerph-20-05295-t004:** Summary of the purpose of fitting joint spatial model, key findings, and reported methodological gaps of selected studies.

Items	Number	Percentage (%)	References
Reasons for using joint modelling (n = 36)			
To borrow strength between diseases and to incorporate data from a more common and related disease when interest is in a relatively rare disease strengthens the relevant results of the rare disease	9	25.00	[36,37,42,44,49,53,55,60,61]
For ease of interpretation, and to improve the precision of estimation	12	33.33	[38,39,43,44,49,53,54,55,57,60,65,76]
To consider the spatial dependence of interrelated outcome variables and to better understand the overlapping epidemiology	14	38.89	[43,44,47,52,57,58,60,61,62,65,72,73,75,78]
To account for such unmeasured exposures that may be common among the diseases	2	5.56	[37,44]
For estimating the relative weight of each shared component for all related disease	6	16.67	[38,41,50,53,65,73]
Key findings			
The joint spatial model yields more precise and efficient estimates especially when the number of desired observed cases is low	6	13.95	[33,43,60,62,73,78]
Found reasonable patterns in the co-occurrence in geographic prevalence across areas	31	72.09	[32,34,35,38,40,41,42,45,46,47,48,49,50,51,52,54,55,56,57,58,59,61,64,65,72,74,75,76,77,79,80]
They had shared risk factors.	7	16.28	[37,39,44,53,60,72,80]
The shared component joint spatial model had a better model fit relative to a joint spatial model without the shared component	5	11.63	[36,46,53,62,73]
Methodological gaps (n = 36)			
A meaningful time period is required to detect the temporal effects	4	11.11	[38,41,44,80]
Assuming the shared and specific components as independent ignores the possibility of interactions between the true covariates	4	11.11	[38,44,62,73]
Edge effects	3	8.33	[36,38,77]
The results are biased by the Modifiable Areal Unit Problem (MAUP)	2	5.56	[48,55]
Aggregation of the data has the effect of introducing ecological fallacy and large geographical units of analysis may mask some information of interest. Results and efficiency may be improved by having smaller units of analysis	7	19.44	[40,47,48,49,53,65,80]
MCMC has a computational problems, model fitting, and convergence issues	3	8.33	[42,43,56]

**Table 5 ijerph-20-05295-t005:** Quality assessment of included studies.

ID	Author	Year	AaO	SaP	MS	MM	PRD	QoD	PoR	IDOR	Sum	Rating
1	Freitas et al., 2022 [75]	2022	2	2	1	1	2	2	1	2	13	High
2	Kazembe et al., 2015 [46]	2015	2	2	1	2	2	2	2	2	15	Very high
3	Kinyoki et al., 2017 [62]	2017	2	2	1	1	2	1	1	2	12	High
4	Besharati et al., 2020 [45]	2020	2	2	1	2	2	1	1	2	13	High
5	Kramer et al., 2013 [48]	2013	1	1	1	1	1	1	1	2	9	Medium
6	Law et al., 2018 [49]	2018	2	2	1	1	2	1	1	2	12	High
7	Lawson et al., 2014 [63]	2014	2	2	2	1	2	0	2	1	12	High
8	Lawson et al., 2020 [51]	2020	1	1	1	1	2	2	1	0	9	Medium
9	Mahaki et al., 2011 [38]	2011	2	2	2	1	2	2	2	1	14	Very high
10	Mahaki et al., 2018 [32]	2018	2	2	1	1	2	1	2	1	12	High
11	Nasrazadani et al., 2018 [39]	2018	2	2	2	1	2	2	2	2	15	Very high
12	Desjardins et al., 2014 [76]	2018	2	2	1	1	2	1	2	1	12	High
13	Odhiambo et al., 2021 [53]	2021	2	2	2	2	2	1	2	2	15	Very high
14	Okango et al., 2015 [54]	2015	2	2	1	1	2	2	1	2	13	High
15	Orunmoluyi et al., 2022 [64]	2022	2	2	1	1	2	2	1	2	13	High
16	Otiende et al., 2020 [65]	2020	2	2	2	2	2	2	2	2	16	Very high
17	Raei et al., 2018 [41]	2018	2	2	1	1	1	1	1	1	10	Medium
18	Ransome et al., 2019 [55]	2019	2	2	1	1	1	2	1	1	11	High
19	Roberts et al., 2020 [75]	2020	2	2	2	2	2	2	2	2	16	Very high
20	Schur et al., 2011 [78]	2011	2	2	1	2	2	2	1	1	13	High
21	Stensgaard et al., 2011 [79]	2011	1	2	2	1	1	1	1	1	10	Medium
22	Stoppa et al., 2022 [40]	2022	1	2	2	2	1	2	1	1	12	High
23	Norwood et al., 2020 [56]	2020	2	2	2	2	2	2	1	1	14	Very high
24	Adebayo et al., 2016 [57]	2016	2	2	2	2	2	2	2	1	15	Very high
25	Asmarian et al., 2019 [42]	2019	2	2	1	2	2	1	1	1	12	High
26	Huang et al., 2018 [58]	2018	2	1	1	2	2	2	2	1	13	High
27	Kang et al., 2014 [59]	2014	2	2	2	2	2	2	2	1	15	Very high
28	Law et al., 2020 [50]	2020	2	2	2	2	2	2	2	1	15	Very high
29	Roberts et al., 2022 [60]	2022	2	2	1	2	2	1	1	1	12	High
30	Carabali et al., 2022 [61]	2022	2	2	2	2	1	2	1	1	13	High
31	Cramb et al., 2015 [37]	2015	2	2	2	2	2	2	1	1	14	Very high
32	Kinyoki et al., 2017 [62]	2017	2	2	1	1	2	1	1	1	11	High
33	Kline et al., 2019 [47]	2019	2	2	1	1	2	2	1	1	12	High
34	Chidumwa et al., 2021 [73]	2021	1	1	1	1	1	1	1	1	8	Medium
35	Adeyemi et al., 2019 [43]	2019	2	2	1	1	2	2	1	1	12	High
36	Darikwa et al., 2019 [74]	2019	2	2	1	1	2	2	1	1	12	High
37	Darikwa et al., 2020 [51]	2020	1	2	1	1	1	1	1	1	9	Medium
38	Chamanpara et al., 2015 [36]	2015	1	1	1	1	1	1	1	1	8	Medium
39	Carroll et al., 2017 [35]	2017	2	2	1	1	1	1	1	1	10	Medium
40	Adegboye et al., 2017 [80]	2017	1	1	1	1	1	1	1	1	8	Medium
41	Neelon et al., 2014 [52]	2014	1	1	1	1	1	1	1	1	8	Medium
42	Ahmadipanahmehrabadi et al., 2019 [33]	2019	2	2	1	2	2	2	1	1	13	High
43	Bermudi et al., 2020 [34]	2020	2	2	2	2	2	2	2	2	16	Very high
Range			1–2	1–2	1–2	1–2	1–2	0–2	1–2	0–2	8–16	
Median score			2	2	1	1	2	2	1	1	12	High
Mean score			1.79	1.84	1.35	1.42	1.72	1.53	1.33	1.28	12.26	

AaO, aims and objectives; SaP, setting and population; MS, model structure; MM, modelling methods; PRDS, parameter ranges and data sources; QoD, quality of data; PoR, presentation of results; IDoR, interpretation, and discussion of results.

## Data Availability

This study is a systematic review of already published articles and the extracted data are provided as a Appendix A.

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
