# Peer review of "A Systematic Review of Joint Spatial and Spatiotemporal Models in Health Research"

_ijerph, 2023, doi:10.3390/ijerph20075295_

Round 1
Reviewer 1 Report
Excellent presentation. All are clear to understand. Some minor comments:
1. The highest number of publications is in Iran. What could be the reason for the highest number?
2. The authors have found 4000 articles from 2011, i.e., about 40 per year. Could this number be significantly lower before 2011? If the number is similar before 2011, say 2000-2010, the study period could be extended (this is not a recommendation); this is an observation; authors may check it.
3. Referencing system is numerical in all Tables except in Table 5. Numbers could be inserted in addition to the first author's name.
4. I think the referencing could be presented with increasing order in Tables as these are in the text (this will look better).
Author Response
The Editors
Application of Statistical Methods in Public Health and Medical Research
08 March 2023
Dear Editors,
Re: Manuscript title: A systematic review of joint spatial and spatiotemporal models in
health research - Submission ID: ijerph-2195131
A point-by-point response to reviewers
We would like to thank the reviewers for their very thoughtful and constructive comments on this manuscript. We have considered all comments and clarifications by editors and reviewers and would like to respond, point-by-point with responses detailed in the following pages. The details of changes are shown by track changes in the supplementary document attached.
Authors and Email
Getayeneh Antehunegn Tesema: getayeneh.tesema@monash.edu
Zemenu Tadesse Tessema: Zemenu.tessema@monash.edu
Stephane Heritier: stephane.heritier@monash.edu
Rob G. Stirling: R.Stirling@alfred.org.au
Arul Earnest: arul.earnest@monash.edu
Reviewer #1
- The highest number of publications is in Iran. What could be the reason for the highest number?
Author's Response: Thank you reviewer for the comment. This might be due to the Iranian government agencies at all levels, municipal, provincial, and national, publishing an abundance of data publicly https://iranopendata.org/en/about, these data are ideal for employing spatial studies. Besides, the majority of the joint spatial studies were on cancer and this might be due to the rapid rise in cancer, and the broad availability of cancer registries, and as you know cancers share common etiology and environmental risk factors that made them suitable for joint spatial analysis.
- The authors have found 4000 articles from 2011, i.e., about 40 per year. Could this number be significantly lower before 2011? If the number is similar before 2011, say 2000-2010, the study period could be extended (this is not a recommendation); this is an observation; authors may check it.
Author's Response: Thank you reviewer for the concern. For our review, the year 2011 was chosen as the starting point as this was the time point from which joint spatial and spatiotemporal analysis of two or more health outcomes became widely implemented as a new area of study. Over the past decade, since 2011, there have been rapid changes in analytical techniques due to the ongoing advances in science and technology. Studies before that date were either outdated or superseded by newer methods included in our review. Before the year 2011, there were very limited published joint spatial and spatiotemporal studies available we have checked this using our search terms. Initially, we did a search since 2000 but very few studies (less than 10 studies were published with the joint spatial and spatiotemporal model) were available before 2011, which is why we decided to consider studies published since 2011.
- Referencing system is numerical in all Tables except in Table 5. Numbers could be inserted in addition to the first author's name.
Author's Response: Thank you, reviewer. We inserted the references in addition to the first Author's name and date. (See Table 5)
- I think the referencing could be presented with increasing order in Tables as these are in the text (this will look better).
Author's Response: Thank you reviewer for the suggestions. The reference is numbered in the order we have cited in the manuscript and therefore as you can see they are cited in increasing order in the body of the manuscript but in the tables they may not be in increasing order because we have used some of the references in the introduction and method section and we used Vancouver referencing that is why the numbers are assigned based on the place the reference is cited.

Reviewer 2 Report
The author overviewed the research lectures with spatial and spatio-temporal models in this manuscript. They are mining the related papers from six electronic and more than 4000 studies. They also summarized these studies and these lectures. The author also summarized the research methods and future trends of this research region. The results and work are very impressive and very interesting. The English writing is very good! However, the scientific contribution is limited and I would like to consider accepting this paper if the authors could include more insights in the results and discussion part from the scientific view.
Author Response
Dear Editors,
Re: Manuscript title: A systematic review of joint spatial and spatiotemporal models in
health research - Submission ID: ijerph-2195131
A point-by-point response to reviewers
We would like to thank the reviewers for their very thoughtful and constructive comments on this manuscript. We have considered all comments and clarifications by editors and reviewers and would like to respond, point-by-point with responses detailed in the following pages. The details of changes are shown by track changes in the supplementary document attached.
Authors and Email
Getayeneh Antehunegn Tesema: getayeneh.tesema@monash.edu
Zemenu Tadesse Tessema: Zemenu.tessema@monash.edu
Stephane Heritier: stephane.heritier@monash.edu
Rob G. Stirling: R.Stirling@alfred.org.au
Arul Earnest: arul.earnest@monash.edu
Reviewer #2
- The author overviewed the research lectures with spatial and spatio-temporal models in this manuscript. They are mining the related papers from six electronic and more than 4000 studies. They also summarized these studies and these lectures. The author also summarized the research methods and future trends of this research region. The results and work are very impressive and very interesting. The English writing is very good! However, the scientific contribution is limited and I would like to consider accepting this paper if the authors could include more insights in the results and discussion part from the scientific view.
Author's Response: Thank you very much, dear reviewer. We included more insights about these systematic review findings and its implication in the discussion section. (See Discussion section, lines 495-511, page 23)

Reviewer 3 Report
This manuscript conducted a systematic review of joint spatial and spatio-temporal models in health care. The topic is very interesting, however, some drawbacks need to be addressed. The major comments of the study are:
First, the reviewer is confused about the definitions of multivariate and joint spatial and spatio-temporal models. The authors should clearly define these terms, especially for joint spatial and spatio-temporal.
Second, the authors selected a bunch of keywords, however, only 43 papers fulfilled the eligibility criteria. Why the authors selected so many keywords? How did authors screen selected papers? Only around 10% of papers were selected in the last, is it reasonable?
Third, the depth and novelty of this manuscript are not enough.
Author Response
The Editors
Application of Statistical Methods in Public Health and Medical Research
08 March 2023
Dear Editors,
Re: Manuscript title: A systematic review of joint spatial and spatiotemporal models in
health research - Submission ID: ijerph-2195131
A point-by-point response to reviewers
We would like to thank the reviewers for their very thoughtful and constructive comments on this manuscript. We have considered all comments and clarifications by editors and reviewers and would like to respond, point-by-point with responses detailed in the following pages. The details of changes are shown by track changes in the supplementary document attached.
Authors and Email
Getayeneh Antehunegn Tesema: getayeneh.tesema@monash.edu
Zemenu Tadesse Tessema: Zemenu.tessema@monash.edu
Stephane Heritier: stephane.heritier@monash.edu
Rob G. Stirling: R.Stirling@alfred.org.au
Arul Earnest: arul.earnest@monash.edu
Reviewer#3
This manuscript conducted a systematic review of joint spatial and spatio-temporal models in health care. The topic is very interesting, however, some drawbacks need to be addressed. The major comments of the study are:
- First, the reviewer is confused about the definitions of multivariate and joint spatial and spatio-temporal models. The authors should clearly define these terms, especially for joint spatial and spatio-temporal.
Author's Response: Thank you reviewer for the concern. We agree with the reviewer, but here we are summarising what other researchers have used in terms of the terminology that has been published and cannot be changed. In our paper, we have kept to a consistent terminology. In our manuscript, the term multivariate and joint spatial and spatiotemporal models are the same. Both are used to define methods that spatially and temporally model two or more health outcomes, or the same health outcome in two or more subsets of the population at risk. Authors used either of the terms for methods employed to analyze spatial and spatiotemporal models applied to two or more health outcomes simultaneously. During searching, some of the studies were published as multivariate spatial and spatiotemporal models and some of the studies used the term joint spatial and spatiotemporal models. We have consistently used joint spatial and spatiotemporal models on our part. However, as this study was a systematic review the authors of the individual papers used joint/multivariate/bivariate spatial models interchangeably to define the spatial modelling of two or more health outcomes simultaneously, that is why in the result section and tables we used these terms as it was presented in the primary paper. Therefore, we had to take a pragmatic approach including both terms in our review definition.
- Second, the authors selected a bunch of keywords, however, only 43 papers fulfilled the eligibility criteria. Why the authors selected so many keywords? How did authors screen selected papers? Only around 10% of papers were selected in the last, is it reasonable?
Author's Response: Thank you reviewer for the comment. We used an array of keywords for searching because there are numerous keywords and synonyms for joint spatial analysis. Different terms are used for joint spatial studies, you can find joint spatial, bivariate spatial, multivariate spatial, geographic, joint spatiotemporal, disease risk mapping, etc mainly to not miss study articles for this review. Searching was done using six databases and then the title and abstract screening were conducted by two reviewers. When conflicts arose we adjudicated through discussion and consulting with a third reviewer based on the inclusion and exclusion criteria. Then full-text screening was done for studies that had to pass the title and abstract screening based on the stated eligibility criteria and conflicts were resolved again using a third-reviewer adjudication process. The screening was done manually, using Endnote and Covidence software. Finally, as you said 43 studies (<10% of the articles) were considered for this review, which is reasonable because during searching using keywords animal studies, non-joint spatial, non-spatial, and wrongly named bivariable and multivariable analysis as bivariate and multivariate spatial models were retrieved but during title and abstract, and full-text screening the articles were extensively evaluated for inclusion based on the eligibility criteria’s that was why only 43 studies were fulfilled the eligibility criteria. Commonly, less than 10% of the retrieved articles are considered for the final review. A similar recent systematic review of Bayesian spatial-temporal models on cancer incidence and mortality similarly identified 38 studies for study evaluation but only 4 (approx. 10%) for study inclusion PMID: 32449006.
- Third, the depth and novelty of this manuscript are not enough.
Author's Response: Thank you reviewer for the comment. We have attempted to address this issue in the discussion section of the manuscript on page 23, lines 495-511, and appended below. ‘‘The scientific community may benefit from the epidemiological and statistical insights this systematic study provides on joint spatial and spatiotemporal model applications in health research. First, the utility of joint spatial and spatiotemporal models is more pronounced in large data registries and when multiple interrelated diseases are fitted simultaneously. It is therefore crucial to estimate the smoothed relative risk of lower prevalence cases through the borrowing of strength from the related cases and neighbourhood areas. Although there were joint spatial and spatiotemporal studies, the systematic review found heterogeneity in methods of estimation technique, statistical models, prior selections, defining adjacencies, and model complexities. This showed that a consistent framework for undertaking joint spatial and spatiotemporal models is needed. This framework is currently a focus of our research program. This systematic review provides insight that jointly modeling two or more cases that have shared characteristics is better to detect clusters of cases specifically when the number of cases is rare such as rare cancer and orphan diseases or the population is small. Another important finding was that the most complex models (joint spatial and spatiotemporal models incorporating covariates and interaction) perform very well. Overall, the systematic review identified several areas of improvement in joint spatial studies such as providing data, maps, scripts and methodological gaps’. (See Discussion section, lines 495-511 and page -23)

Reviewer 4 Report
Please see attachement

Author Response
The Editors
Application of Statistical Methods in Public Health and Medical Research
08 March 2023
Dear Editors,
Re: Manuscript title: A systematic review of joint spatial and spatiotemporal models in
health research - Submission ID: ijerph-2195131
A point-by-point response to reviewers
We would like to thank the reviewers for their very thoughtful and constructive comments on this manuscript. We have considered all comments and clarifications by editors and reviewers and would like to respond, point-by-point with responses detailed in the following pages. The details of changes are shown by track changes in the supplementary document attached.
Authors and Email
Getayeneh Antehunegn Tesema: getayeneh.tesema@monash.edu
Zemenu Tadesse Tessema: Zemenu.tessema@monash.edu
Stephane Heritier: stephane.heritier@monash.edu
Rob G. Stirling: R.Stirling@alfred.org.au
Arul Earnest: arul.earnest@monash.edu
Reviewer#4
- This work engaged a systematic review of joint spatial and spatiotemporal methods in health research. Mainly applied to infectious diseases, cancer, chronic diseases, and maternal and child health outcomes. This shows that infectious diseases, which have a major worldwide burden and have the potential to spread to nearby areas, are currently receiving significant attention from researchers. The spatial and spatio-temporal study of interrelated infectious diseases provides a better understanding of the magnitude, pattern, and individual and area-level risk factors of infectious diseases. This review found that when the outcome is rare, joint spatial and spatiotemporal models have proven effective because they improve statistical power by borrowing strength from related health outcomes. Therefore, this systematic review highlighted the need for future joint spatial and Spatio-temporal models to analyse correlated health outcomes to guide decision-making for effective prevention and control strategies. I think this work is interesting and deserves to be published in International Journal of Environmental Research and Public Health.
Author's response: We thank you for your comments.

Round 2
Reviewer 2 Report
The author did add some insights in the discussion part. I would like to suggest editor to accept this manuscript for publication.
Reviewer 3 Report
The authors addressed all comments. Ready for publication.